# Learning to Reason: Distilling Hierarchy via Self-Supervision and Reinforcement Learning

## Abstract

We present a hierarchical planning and control framework that enables an agent to perform various tasks and adapt to a new task flexibly. Rather than learning an individual policy for each particular task, the proposed framework, DISH, distills a hierarchical policy from a set of tasks by self-supervision and reinforcement learning. The framework is based on the idea of latent variable models that represent high-dimensional observations using low-dimensional latent variables. The resulting policy consists of two levels of hierarchy: (i) a planning module that *reasons* a sequence of latent intentions that would lead to optimistic future and (ii) a feedback control policy, shared across the tasks, that *executes* the inferred intention. Because the reasoning is performed in low-dimensional latent space, the learned policy can immediately be used to solve or adapt to new tasks without additional training. We demonstrate the proposed framework can learn compact representations (3- and 1-dimensional latent states and commands for a humanoid with 197- and 36-dimensional state features and actions) while solving a small number of imitation tasks, and the resulting policy is directly applicable to other types of tasks, i.e., navigation in cluttered environments. The supplementary video is available at: https://bit.ly/2rwIfQn

## 1 Introduction

Reinforcement learning (RL) aims to compute the optimal control policy while an agent interacts with the environment. Recent advances in deep learning enable RL frameworks to utilize deep neural networks to efficiently represent and learn a policy having a flexible and expressive structure. As a result, we've been witnessing RL agents that already achieved or even exceeded human-level performances in particular tasks (Mnih et al., 2015; Silver et al., 2017). The core of intelligence, however, is not just to learn a policy for a particular problem instance, but to solve various multiple tasks or immediately adapt to a new task. Given that a huge computational burden makes it unrealistic to learn an individual policy for each task, an agent should be able to *reason* about its action. If predictions about consequences of actions are available, e.g., by using an internal model (Ha & Schmidhuber, 2018; Kaiser et al., 2019), an intelligent agent can plan a sequence of its actions. Involving planning procedures in a control policy could provide adaptiveness to an agent, but it is often not trivial to learn such a prediction & planning framework: First, it is difficult to obtain the exact internal dynamic model directly represented in high-dimensional state (observation) space. Model errors inevitably become larger in the high-dimensional space and are accumulated along the prediction/planning horizon. This prohibits planning methods from producing a valid prediction and so a sensible plan. Second, and perhaps more importantly, planning methods cannot help but relying on some dynamic programming or search procedures, which quickly become intractable for problems with high degrees of freedom (DOFs) because the size of search space grows exponentially with DOFs, i.e., the curse of dimensionality (LaValle, 2006).

Crucial evidence found in the cognitive science field is that there exists a certain type of hierarchical structure in the humans' motor control scheme addressing the aforementioned fundamental difficulty (Todorov & Ghahramani, 2003; Todorov, 2004). Such a hierarchical structure is known to utilize two levels of parallel control loops, operating in different time scales; in a coarser scale, the high-level loop generates task-relevant commands for the agent to perform a given task, and then in a finer time scale, the low-level loop maps those commands into control signals while actively reacting to disturbances that the high-level loop could not consider (e.g., the spinal cord) (Todorov

& Ghahramani, 2003). Because the low-level loop does not passively generate control signals from high-level commands, the high-level loop is able to focus only on the task-relevant aspects of the environment dynamics that can be represented in a low-dimensional form. Consequently, this hierarchical structure allows us for efficiently predicting and planning the future states to compute the commands.

Motivated by this evidence, we propose a framework, termed "DISH", that DIStills a Hierarchical structure for reasoning and control. As depicted in Fig. 1, the proposed framework has two levels of hierarchy. The high-level loop represents an agent's current state as a low-dimensional latent state and generates/reasons task-relevant high-level commands by predicting and planning the future in the latent space. The low-level loop receives the high-level commands as well as the current states and maps them into the high-dimensional control signal. Two different types of learning are required to build such a framework: (i) a low-dimensional latent representation for an internal model should be obtained from agent's own experiences via *self-supervised learning*; (ii) a control policy should be learned while interacting with the environment via *reinforcement learning*.

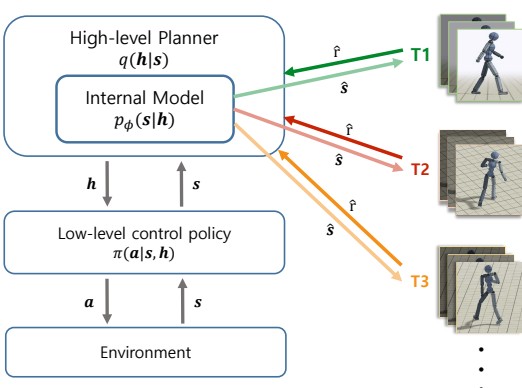

Figure 1: The DISH framework.

We combined these two learning problems by transforming a multitask RL problem into generative model learning using the control-inference duality (Levine, 2018; Todorov, 2008; Rawlik et al., 2012). In this perspective, an agent equipped with a low-level control policy is viewed as a generative model that outputs trajectories according to high-level commands. Reasoning the high-level commands is then considered as a posterior inference problem; we introduce a low-dimensional internal model to make this inference tractable. We demonstrate that the proposed framework can learn the compact representation (3-dimensional latent states for a humanoid robot having 90-dimensional states) and the control policy while solving a small number of imitation tasks, and the learned planning and control scheme is immediately applicable to new tasks, e.g., navigation through a cluttered environment.

## 2 RELATED WORK

***Hierarchical RL:*** To apply task-specific policies learned from individual RL problems to various tasks, hierarchical structures are often considered where each learned policy serves as a low-level controlller, i.e., as a "skill", and a high-level controller selects which skills to perform in the context the agent lies at (Peng et al., 2018; 2019; Merel et al., 2019a; Lee et al., 2019). Peng et al. (2018; 2019) trained robust control policies for imitating a broad range of example motion clips and integrated multiple skills into a composite policy capable of executing various tasks. Merel et al. (2019a) similarly trained many imitation policies and utilized them as individual skills that a high-level controller chooses based on the visual inputs. Lee et al. (2019) included transition policies which help the agent smoothly switch between the skills. Another line of approaches is using continuous-valued *latent* variables to represent skills (Co-Reyes et al., 2018; Gupta et al., 2018; Eysenbach et al., 2019; Florensa et al., 2017; Hausman et al., 2018). Co-Reyes et al. (2018) proposed an autoencoder-like framework where an encoder compresses trajectories into latent variables, a state decoder reconstructs trajectories, and a policy decoder provides a control policy to follow the reconstructed trajectory. Gupta et al. (2018); Eysenbach et al. (2019); Florensa et al. (2017) also introduced latent variables to efficiently represent various policies. Instead of using one *static* latent variable, Merel et al. (2019b) proposed a framework that encodes expert's demonstrations as latent *trajectories* and infers a latent trajectory from an unseen skill for one-shot imitation. Haarnoja et al. (2018a) proposed a hierarchical structure for RL problems where marginalization of low-level actions provides a new system for high-level action. In their framework, policies at all levels can be learned with different reward functions such that a high-level policy becomes easier to be optimized from the marginalization.

Note that the above hierarchical RL approaches train the high-level policy by solving another RL problem; because the individual skill or the latent variables compress dynamics of the agent,

variations of them provide efficient exploration for the high-level RL. Our framework also considers low-dimensional and continuous latent *trajectories* to represent various policies. Rather than learning a high-level policy, however, our framework learns an internal model with which the high-level module performs reasoning; the agent can efficiently reason its high-level commands by searching the low-dimensional latent space with the learned internal model. The learned planning/control structure is then directly applicable to new sets of tasks the agent hasn't met during training. Only a few recent works (Hafner et al., 2019; Sharma et al., 2019) incorporated reasoning processes into high-level modules, but neither of them exploits low-dimensional latent space for planning (Sharma et al., 2019) nor low-dimensional commands (Hafner et al., 2019). Our ablation study in Section 4.1 shows the effectiveness of utilizing both latent states and commands and, to our best knowledge, DISH is the first framework doing so.

***Model-based RL & Learning to Plan:*** Model-based RL algorithms attempt to learn the agent's dynamics and utilize the planning and control methods to perform tasks (Williams et al., 2017; Deisenroth et al., 2015; Chua et al., 2018). Williams et al. (2017); Chua et al. (2018) utilized deep neural networks to model the dynamics and adopted the model predictive control method on the learned dynamics; Deisenroth et al. (2015) used the Gaussian processes as system dynamics, which leads to the efficient and stable policy search. Though these methods have shown impressive results, they are not directly applicable to systems having high DOFs because high-dimensional modeling is hard to be exact and even advanced planning and control methods are not very scalable to such systems. One exceptional work was proposed by Ha & Schmidhuber (2018), where the variational autoencoder and the recurrent neural network are combined to model the dynamics of the observation. They showed that a simple linear policy w.r.t the low-dimensional latent state can control the low DOFs agent, but (i) high-DOFs systems require a more complicated policy structure to output high-dimensional actions and (ii) reasoning (or planning) by predicting the future is essential to solve a set of complex tasks. On the other hand, Ha et al. (2018a;b) trained the low-dimensional latent dynamics from expert's demonstrations and generated motion plans using the learned dynamics; the high-dimensional motion plans were able to be computed efficiently, but the control policy for executing those plans was not considered. Some recent works have attempted to build the policy network in such way that resembles the advanced planning and optimal control methods: Tamar et al. (2016) encoded the value iteration procedures into the network; Okada et al. (2017); Amos et al. (2018) wired the network so as to resemble the path-integral control and the iterative LQR methods, respectively. The whole policy networks are trained end-to-end and, interestingly, system dynamics and a cost function emerge during the learning procedure. However, these methods were basically designed just to mimic the expert's behaviors, i.e., addressing inverse RL problems, and also tried to find the control policy directly in the (possibly high-dimensional) state space.

## 3 DISH: DISTILLING HIERARCHY FOR PLANNING AND CONTROL

### 3.1 MULTITASK RL AS LATENT VARIABLE MODEL LEARNING

Suppose that a dynamical system with states $\mathbf{s} \in \mathcal{S}$ is controlled by actions $\mathbf{a} \in \mathcal{A}$, where the states evolve with the stochastic dynamics $p(\mathbf{s}_{k+1}|\mathbf{s}_k, \mathbf{a}_k)$ from the initial states $p(\mathbf{s}_1)$. Let $\tilde{r}_k(\mathbf{s}_k, \mathbf{a}_k)$ denote a reward function that the agent wants to maximize with the control policy $\pi_\theta(\mathbf{a}_k|\mathbf{s}_k)$. Reinforcement learning problems are then formulated as the following optimization problem:

$$\theta^* = \arg\max_\theta \mathbb{E}_{q_\theta(\mathbf{s}_{1:K}, \mathbf{a}_{1:K})} \left[ \sum_{k=1}^K \tilde{r}_k(\mathbf{s}_k, \mathbf{a}_k) \right], \tag{1}$$

where the *controlled* trajectory distribution $q_\theta$ is given by:

$$q_\theta(\mathbf{s}_{1:K}, \mathbf{a}_{1:K}) \equiv p(\mathbf{s}_1) \prod_{k=1}^K p(\mathbf{s}_{k+1}|\mathbf{s}_k, \mathbf{a}_k)\pi_\theta(\mathbf{a}_k|\mathbf{s}_k). \tag{2}$$

By introducing an artificial binary random variable $o_t$, called the *optimality variable*, whose emission probability is given by exponential of a state-dependent reward, i.e. $p(O_k = 1|\mathbf{s}_k) = \exp(r_k(\mathbf{s}_k))$, and by defining an appropriate action prior $p(\mathbf{a})$ and corresponding the *uncontrolled* trajectory distribution, $p(\mathbf{s}_{1:K}, \mathbf{a}_{1:K}) \equiv p(\mathbf{s}_1) \prod_{k=1}^K p(\mathbf{s}_{k+1}|\mathbf{s}_k, \mathbf{a}_k)p(\mathbf{a}_k)$, we can view the above RL problem as a probabilistic inference problem for a graphical model in Fig 2(a). The objective of such an

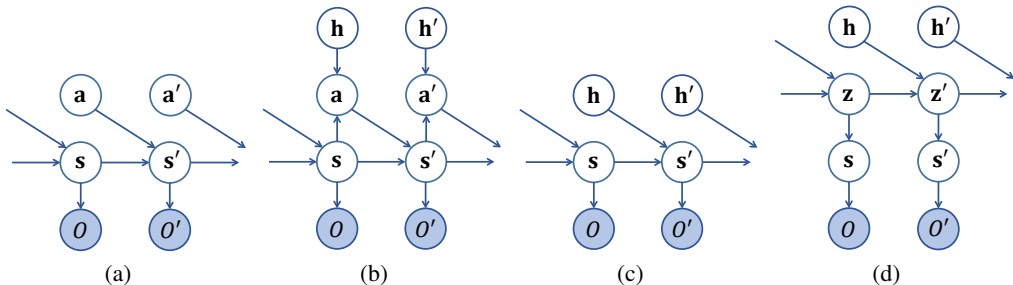

Figure 2: (a) The conventional RL and (b) the proposed hierarchical RL framework. (c) The action-marginalized inference problem. (d) A low-dimensional LVM for the high-level reasoning.

inference problem is to find the optimal variational parameter, $\theta$, such that the controlled trajectory distribution $q_\theta(\mathbf{s}_{1:K}, \mathbf{a}_{1:K})$ fits the posterior distribution $p(\mathbf{s}_{1:K}, \mathbf{a}_{1:K}|O_{1:K} = 1)$ best. More detailed derivations of this duality can be found in Appendix A.2 or in the tutorial paper (Levine, 2018).

Rather than solving one particular task, i.e., one reward function, agents are often required to perform various tasks. Let $\mathcal{T}$ be a set of tasks, and $\pi_{\theta_t^*}(\mathbf{a}_k|\mathbf{s}_k)$ be the optimal policy for $t^{\text{th}}$ task, i.e.,

$$\theta_t^* = \arg\max_{\theta_t} \mathbb{E}_{q_{\theta_t}(\mathbf{s}_{1:K}, \mathbf{a}_{1:K})} \left[ \sum_{k=1}^{K} \tilde{r}_k^{(t)}(\mathbf{s}_k, \mathbf{a}_k) \right], \ \forall t \in \mathcal{T}. \tag{3}$$

For high DOF systems, where policies $\pi_{\theta_t}$ represent a mapping from a high-dimensional state space to a high-dimensional action space, individually optimizing each policy is computationally too expensive. Instead of doing so, we can assume that tasks the agent needs to perform require similar solution properties and consequently the optimal policies have some sort of common structures. We can then introduce a low-dimensional latent variable $\mathbf{h}^{(t)}$ that, by compressing a particular aspect of $\pi_{\theta_t}$ over all the policies, each policy can be conditioned on as $\pi_\theta(\mathbf{a}_k|\mathbf{s}_k, \mathbf{h}^{(t)})$.

Such a hierarchical structure is depicted as Fig. 2(b), where $\mathbf{h}$ can be interpreted as high-level *commands*. We can then define the uncontrolled and the task $t$'s controlled trajectory distributions as

$$p(\mathbf{s}_{1:K}, \mathbf{a}_{1:K}, \mathbf{h}_{1:K}) \equiv p(\mathbf{s}_1) \prod_{k=1}^{K} p(\mathbf{s}_{k+1}|\mathbf{s}_k, \mathbf{a}_k)p(\mathbf{a}_k)p(\mathbf{h}_k), \tag{4}$$

$$q_\theta^{(t)}(\mathbf{s}_{1:K}, \mathbf{a}_{1:K}, \mathbf{h}_{1:K}) \equiv p(\mathbf{s}_1) \prod_{k=1}^{K} p(\mathbf{s}_{k+1}|\mathbf{s}_k, \mathbf{a}_k)\pi_\theta(\mathbf{a}_k|\mathbf{s}_k, \mathbf{h}_k)q^{(t)}(\mathbf{h}_k|\mathbf{s}_k), \tag{5}$$

receptively. In other words, the control policy $\pi_\theta$ is shared across all the tasks, actively mapping high-level commands $\mathbf{h}$, into actual actions, $\mathbf{a}$. Only high-level commands vary with the given task specifications. In the perspective of *control as inference*, a corresponding inference problem now has two parts: one for the policy parameter $\theta$ and another for the task-specific commands $\mathbf{h}$. Note that, if a high-level policy $\bar{\bar{\pi}}_\theta(\mathbf{h}|\mathbf{s})$ is used to compute high-level commands, the learning problem then becomes the standard Hierarchical RL (HRL). We instead introduce a reasoning module to generate high-level commands which infers the optimal $\mathbf{h}$ for a given task $t$ and a current state $\mathbf{s}$ by predicting futures. As often used in many HRL methods, the high-level module of the proposed framework operates in a coarser time scale than the low-level policy does.

Similar to the latent model learning in Appendix A.3 and the control-inference duality in Appendix A.2, we can derive the following lower-bound of optimality likelihood $\mathcal{L}^{(t)}$ for a task $t$:

$$\log p_\theta(O_{1:K}^{(t)} = 1) = \log \int p(O_{1:K}^{(t)} = 1|\mathbf{s}_{1:K})p(\tau)\frac{q_\theta^{(t)}(\tau)}{q_\theta^{(t)}(\tau)}d\tau$$

$$\geq \mathbb{E}_{q_\theta^{(t)}(\tau)} \left[ \sum_{k=1}^{K} r_k^{(t)}(\mathbf{s}_k) - \log \frac{\pi_\theta(\mathbf{a}_k|\mathbf{s}_k, \mathbf{h}_k)}{p(\mathbf{a}_k)} - \log \frac{q^{(t)}(\mathbf{h}_k|\mathbf{s}_k)}{p(\mathbf{h}_k)} \right] \equiv \mathcal{L}^{(t)}(\theta, q), \tag{6}$$

where $\tau \equiv (\mathbf{s}_{1:K}, \mathbf{a}_{1:K}, \mathbf{h}_{1:K})$. This suggests a novel learning scheme of the hierarchical policy in Equation 5: (i) For a given task $t$ and a fixed low-level policy $\pi_\theta$, high-level commands $\mathbf{h}_k$ are computed via variational inference. This inference procedure $q(\mathbf{h}|\mathbf{s})$ should take predictions about future rewards into account to generate $\mathbf{h}$, which can be interpreted as planning. To do so, we build an internal model via self-supervised learning and perform planning with the internal model. (ii) With the planning module equipped, a low-level policy $\pi_\theta(\mathbf{a}|\mathbf{s}, \mathbf{h})$ generates control actions $\mathbf{a}$ as in RL problems, which can be trained using standard deep RL algorithms (Schulman et al., 2017; Haarnoja et al., 2018b).

## 3.2 SELF-SUPERVISED LEARNING OF INTERNAL MODEL

The role of $q(\mathbf{h}|\mathbf{s})$ is to compute the high-level commands that would lead to maximum accumulated rewards in the future; as shown in Equation 6, this infers the commands that maximizes the likelihood of optimality variables when $O_{1:K} = 1$ were observed. Given that the ELBO gap is the KL-divergence between the posterior and variational distributions, it is obvious that more exact variational inference will make the lower bound tighter, thereby directly leading to the agent's better performance as well as the better policy learning. What would the exact posterior be like? Fig. 2(c) shows the graphical model of the inference problem that $q(\mathbf{h}|\mathbf{s})$ should address, which is obtained by marginalizing actions from Fig. 2(b); as also shown in (Haarnoja et al., 2018a), such marginalization results in a new system with new control input $\mathbf{h}$, thus the inference problem in this level is again the RL/OC problem. To get the command at the moment, $\mathbf{h}_1$, the inference procedure should compute the posterior command trajectories $\mathbf{h}_{1:K}$ by considering the dynamics and observations (the optimality variables), and marginalize the future commands $\mathbf{h}_{2:K}$ out. Though the dimensionality of $\mathbf{h}$ is much lower than that of $\mathbf{a}$, this inference problem is still not trivial to solve by two reasons: (i) The dynamics of states $p_\theta(\mathbf{s}'|\mathbf{s}, \mathbf{h}) = \int p(\mathbf{s}'|\mathbf{s}, \mathbf{a})\pi_\theta(\mathbf{a}|\mathbf{s}, \mathbf{h})d\mathbf{a}$ contains the environment component of which information can be obtained only through expensive interactions with the environment. (ii) One might consider building a surrogate model $p_\phi(\mathbf{s}'|\mathbf{s}, \mathbf{h})$ via supervised learning with transition data obtained during low-level policy learning and utilizing the learned model for inference. However, learning high-dimensional transition model is hard to be accurate and the inference (planning) in high-dimensional space is intractable because of, e.g., the curse of dimensionality (Ha et al., 2018a).

However, we can reasonably assume that configurations that should be considered from planning form some sort of low-dimensional manifold in the original space (Vernaza & Lee, 2012), and the closed-loop system with high-level commands provides stochastic dynamics on that manifold. That is, a high-dimensional transition model in Fig. 2(c) can be represented as a latent variable model (LVM) in Fig. 2(d). Once this low-dimensional representation is obtained, any motion planning or inference algorithms can solve the variational inference problem very efficiently with the vastly restricted search space.

Our framework collects the trajectories from low-level policies and utilize them to learn a LVM for inference, which is formulated as a maximum likelihood estimation (MLE) problem. Suppose that we have collected a set of state trajectories and latent commands $\{\mathbf{s}_{1:K}^{(n)}, \mathbf{h}_{1:K}^{(n)}\}_{n=1,...,N}$. We then formulate the MLE problem as:

$$\phi^* = \arg\max_\phi \sum_n \log p_\phi(\mathbf{s}_{1:K}^{(n)}|\mathbf{h}_{1:K}^{(n)}). \tag{7}$$

As in Fig. 2(d), the states are assumed to be emerged frwwwwwwwwwwom a latent dynamical system, where a latent state trajectory, $\mathbf{z}_{1:K}$, lies on a low-dimensional latent space $\mathcal{Z}$:

$$p_\phi(\mathbf{s}_{1:K}|\mathbf{h}_{1:K}) = \int p_\phi(\mathbf{s}_{1:K}|\mathbf{z}_{1:K})p_\phi(\mathbf{z}_{1:K}|\mathbf{h}_{1:K})d\mathbf{z}_{1:K}. \tag{8}$$

In particular, we consider the state space model where latent states follow stochastic transition dynamics with $\mathbf{h}$ as inputs, i.e., the prior $p_\phi(\mathbf{z}_{1:K}|\mathbf{h}_{1:K})$ is a probability measure of a following system:

$$\mathbf{z}_{k+1} = f_\phi(\mathbf{z}_k) + \sigma_\phi(\mathbf{z}_k)(\mathbf{h}_k + \mathbf{w}_k), \ \mathbf{w}_k \sim \mathcal{N}(0, I) \tag{9}$$

and also a conditional likelihood of a state trajectory is assumed to be factorized along the time axis as: $\mathbf{s}_k \sim \mathcal{N}(\mu_\phi(\mathbf{z}_k), \Sigma_\phi(\mathbf{z}_k)) \ \forall k$. The resulting sequence modeling is a self-supervised learning problem that has been extensively studied recently (Karl et al., 2017; Krishnan et al., 2017; Fraccaro

et al., 2017; Ha et al., 2018b). In particular, we adopt the idea of Adaptive path-integral autoencoder in (Ha et al., 2018b), where the variational distribution is parameterized by the controls, $\mathbf{u}$, and an initial distribution, $q_0$, i.e., the proposal $q_{\mathbf{u}}(\mathbf{z}_{[0,T]})$ is a probability measure of a following system:

$$\mathbf{z}_{k+1} = f_\phi(\mathbf{z}_k) + \sigma_\phi(\mathbf{z}_k)\left(\mathbf{h}_k + \mathbf{u}_k + \mathbf{w}_k\right), \ \mathbf{w}_k \sim \mathcal{N}(0, I). \tag{10}$$

Compared to the original formulation in (Ha et al., 2018b), the probability model here is conditioned on the commands, $\mathbf{h}_{1:K}$, making the learning problem conditional generative model learning (Sohn et al., 2015).[1] Note that it is also possible to first obtain a low-dimensional representation considering each state (not trajectory) independently and then fit their dynamics using RNNs like World Model (Ha & Schmidhuber, 2018), or to stack two consecutive observations and learn the dynamical model considering the stacked data as one observation like E2C (Watter et al., 2015). However, Ha et al. (2018b) showed that the representations learned from the short horizon data easily fail to extract enough temporal information and a latent dynamical model suitable for planning can be well-obtained only when considering long trajectories.

### 3.3 REASONING (PLANNING) WITH LEARNED INTERNAL MODEL

Once the LVM is trained, a planning module can efficiently explore the state space $\mathcal{S}$ through the latent state $\mathbf{z}$ and infer the latent commands $\mathbf{h}_{1:K}$ that are likely to result in high rewards; in particular, we adopt a simple particle filter algorithm for inference, because it is known to perform well with non-linear and non-Gaussian systems (Ha et al., 2018a; Piche et al., 2019). Particle filtering, which is also called the sequential Monte-Carlo, utilizes a set of samples and their weights to represent a posterior trajectory distribution. Starting from the initial state, it propagates a set of samples according to the dynamics (Equation 9) and updates the weights using the observation likelihood as $w' \propto w \times p(O_k = 1|\mathbf{s}_k)$. It also resamples the low-weighted particles to maintain the effective sample size. In the perspective of this work, this procedure can be viewed as the agent simulating multiple future state trajectories with the internal model, assigning each of them according to the reward, and reasoning the command that leads to the best-possible future. The detailed explanation is elaborated in Appendix A.4 and in Algorithm 2. Note that for the more complex environment, we can also iterate the whole procedure multiple times to compute a better command, then the planning algorithm becomes the adaptive path integral method (Kappen & Ruiz, 2016; Williams et al., 2017; Ha et al., 2018b). If the resampling procedure is eliminated, it is equivalent to the widely-used cross entropy method (Hafner et al., 2019). Any other inference/planning algorithms compatible with the graphical model of Fig. 2(d) can be also used.

### 3.4 MAIN LEARNING ALGORITHM

---
**Algorithm 1** DIStilling Hierarchy for Planning and Control
---
1: Initialize policy $\theta$ and latent model $\phi$
2: **for** $l = 1, ..., L$ **do**
3:     **while** not converged **do**
4:         Sample a task $t \in \mathcal{T}$
5:         Run the policy $\mathbf{a} \sim \pi_\theta(\mathbf{a}|\mathbf{s}, \mathbf{h})$ with high-level commands $\mathbf{h} \sim q_\phi(\mathbf{h}|\mathbf{s}; t)$
6:         Store trajectories $\tau$ into the experience buffer
7:         Train the policy $\pi_\theta$ using e.g. PPO         ▷ Equation 6
8:     **end while**
9:     Random sample $\mathbf{h}$ and collect rollouts.
10:     Train the internal model using e.g. APIAE         ▷ Equation 7
11: **end for**
---

The overall learning procedure is summarized in Algorithm 1. The procedure consists of an outer internal model learning loop and an inner policy update loop. During the policy update stage (inner loop), the algorithm samples a task, solves the sampled task by using the hierarchical policy, and collects trajectories into the experience buffer. At each time step, the low-level policy decides actions the agent takes under the high-level commands determined by the planning module equipped with the

---

[1]Effectively it only shifts the control input prior from $\mathcal{N}(\mathbf{0}, I)$ to $\mathcal{N}(\mathbf{h}, I)$ as written in Equation 9 and Equation 10 (Williams et al., 2017).

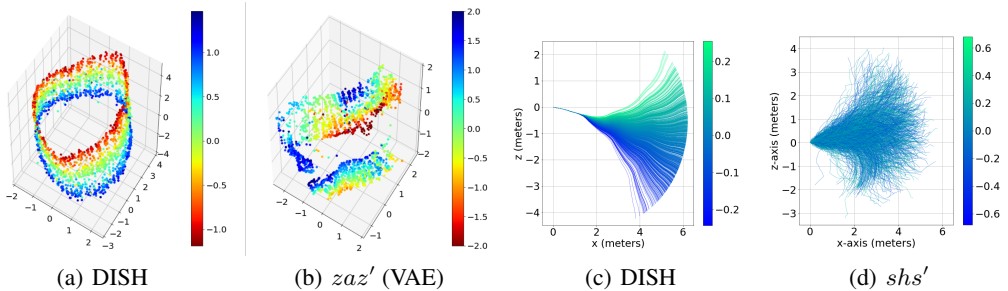

| (a) DISH | (b) $zaz'$ (VAE) | (c) DISH | (d) $shs'$ |

Figure 3: (a), (b) Learned latent models colored by angular velocity. (c), (d) Rollout samples in horizontal (x-z plane) colored by latent command value.

internal latent model. Using trajectory data in the buffer, the low-level policy is updated via a deep RL algorithm (e.g., policy gradient methods). After the low-level policy update, DISH collects another rollouts by random sampling a latent variable **h**, and the internal model is learned via self-supervised learning. These two learning procedures are then iterated for $L$ times.

Note that, for complex systems, tasks can be selected more carefully (at line 4) for the better learning landscape; for example, in earlier phases where the agent couldn't yet learn a valid policy and/or an internal model, the agent can first learn them through imiation learning of expert's demonstrations (Peng et al., 2018) or play data (Lynch et al., 2019) or through intrinsic motivations to acquire useful skills by itself (Sharma et al., 2019). As more challenging tasks are gradually provided to the agent, the internal model (or the reasoning module) is learned to cover wider ranges of state space for those tasks and the low-level policy is trained such that it can be operated with more complicated high-level commands.

## 4 EXPERIMENT

In this section, we demonstrate the effectiveness of the proposed framework on performing planning and control for the high dimensional humanoid model (Peng et al., 2018) which has 197 state features and 36 action parameters, simulated by 1.2kHz Bullet physics engine (Coumans et al., 2013). The low-level control policy and the internal latent model are trained through the imitation learning, where three locomotion data from the Carnegie Mellon University motion capture (CMU mocap) database are used as target motions of imitation. The control policy is trained with the DeepMimic imitation reward (Peng et al., 2018) by using proximal policy optimization (PPO) (Schulman et al., 2017), while the internal model is learned to maximize the likelihood of experience data (i.e. Equation 7) by using the APIAE approach (Ha et al., 2018b). The internal model of DISH is constructed to have a 3-dimensional latent state and a 1-dimensional latent command for all experiments. The low-level policy and the internal model are operated in different time scales, 30Hz and 1Hz, respectively. The learned hierarchical model is then evaluated on trajectory following and navigation tasks in Section 4.1 and 4.2, respectively. For planning and execution, the model predictive control (MPC) scheme with particle filtering (A.4) is used; a 5-second trajectory is planned and the first reasoned high-level command is applied to the low-level policy at 1Hz and 4Hz for each task.

We refer to the appendix for the reward functions, hyperparmeters, and network architectures (A.5 and A.6), task configurations (A.7), and more experimental results (A.8). Our TensorFlow (Abadi et al., 2015) implementation will be made available in the final manuscript. The videos of the training procedure and the resulting policy are available at: https://bit.ly/2rwIfQn

### 4.1 ABLATION STUDY: LEARNING HIERARCHICAL STRUCTURE

In the first experiment, we examine how effectively the proposed framework learns and exploits the internal model. To investigate the effectiveness of each component introduced, we conduct ablation studies by considering three baselines: (i) $sas'$ that does not have neither the hierarchical structure nor LVMs (Fig. 2(a)), (ii) $shs'$ that utilizes the hierarchical policy but doesn't learn the low-dimensional latent dynamics (Fig. 2(c)), and (iii) $zaz'$ that considers the latent dynamics but without

Table 1: Comparison between different types of internal models.

|  | command, $\mathbf{h}$ | LVM, $\mathbf{z}$ | Algorithm |
|---|---|---|---|
| $zhz'$ | O | O | DISH (ours) |
| $sas'$ | X | X | Tamar et al. (2016); Okada et al. (2017); Amos et al. (2018) |
| $shs'$ | O | X | Sharma et al. (2019) |
| $zaz'$ | X | O | Ha & Schmidhuber (2018); Hafner et al. (2019) |

Table 2: Quantitative comparison for trajectory following tasks. 'F' denotes that it was not able to record the true trajectory since the agent kept falling.

|  | Reconstruction | ‖ref-true‖ | ‖plan-ref‖ | ‖plan-true‖ |
|---|---|---|---|---|
| DISH (ours, L=1, Fig. 1 & Fig.2(d)) | 0.3820 | 0.1638 | 0.1576 | **0.0930** |
| DISH+ (ours, L=2) | 0.8414 | **0.1452** | **0.1509** | 0.1105 |
| $sas'$ (w/o command & LVM, Fig. 2(a)) | **0.1289** | F | 0.1771 | F |
| $shs'$ (w/o LVM, Fig. 2(c)) | 0.1393 | 0.2226 | 0.1579 | 0.2231 |
| $zaz'$ (w/o command) | 2.3351 | F | 0.2589 | F |

the hierarchical structure (no latent commands, a LVM version of Fig. 2(a)[2]). Given the rollouts $\{\tau^{(i)}\} = \{\mathbf{s}_{1:K}^{(i)}, \mathbf{a}_{1:K}^{(i)}, \mathbf{h}_{1:K}^{(i)}\}$, learning $sas'$ and $shs'$ are simply supervised learning problems. For the $zaz'$ model, the variational autoencoder (VAE) approach (Kingma & Welling, 2013) is taken to train mappings between the observation and the latent space, and then the latent dynamics is trained via supervised learning, following the idea of (Ha & Schmidhuber, 2018). Note that most HRL frameworks can be categorized as either $zaz'$ e.g., (Ha & Schmidhuber, 2018; Hafner et al., 2019) or $shs'$ e.g., (Sharma et al., 2019). The similar network structures are used for the baselines; implementation details of the baselines also can be found in A.6. Table 1 summarizes the different features of the models with the related works.

Figs. 3(a) and 3(b) show the learned latent space colored by the moving-averaged angular velocity of the ground truth motion. In the case of DISH, the latent state forms a manifold of a cylindrical shape in 3-dimensional space where the locomotion phase and the angular velocity are well encoded along the manifold. In contrast, the latent state structure of the $zaz'$ model does not capture the phase information and failed to construct a periodic manifold, which prevents a valid latent dynamics from being learned. Figs. 3(c) and 3(d) show the rollout trajectories from each internal model colored by the values of high-level commands, $\mathbf{h}$. The high-level commands of DISH are learned to control the heading direction of the humanoid so that the agent can make the structural exploration in the configuration space. The $shs'$ model, on the other hand, fails to learn a valid controlled dynamics (since its space is too large) and consequently just generates noisy trajectories.

To quantitatively evaluate the reasoning performance of DISH and its ability to flexibly perform different tasks, we compare DISH to the baseline models on three trajectory following tasks: going straight, turning left and right. Table 2 reports the RMS errors for reconstruction and differences between the reference, planned, and executed trajectories. There are three things we can observe from the table: (i) Although $sas'$ has the lowest reconstruction error, the computed action from its internal model even cannot make the humanoid walk. This is because the humanoid has a highly unstable dynamics and reasoning of the high-dimensional action is not accurate enough to stabilize the humanoid dynamics, i.e., searching over the 36-dimensional action space with the limited number of particles (1024 in this case) is not feasible. For the same reason, $zaz'$ also fails to let the humanoid walk. (ii) Only the models considering the hierarchical policies can make the humanoid walk, and the DISH framework generates the most executable and valuable plans; the humanoid with the $shs'$ model walks just in random directions rather than following a planned trajectory (see Fig. 3(d)), which implies that the high-level command $\mathbf{h}$ does not provide any useful information regarding the navigation. (iii) By iterating the low-level policy and the internal model learning further, DISH+ becomes able to reason better plans as well as execute them better. Further analysis can be found in A.8

---

[2]Note that Fig.2(d) depicts a LVM version of Fig. 2(c).

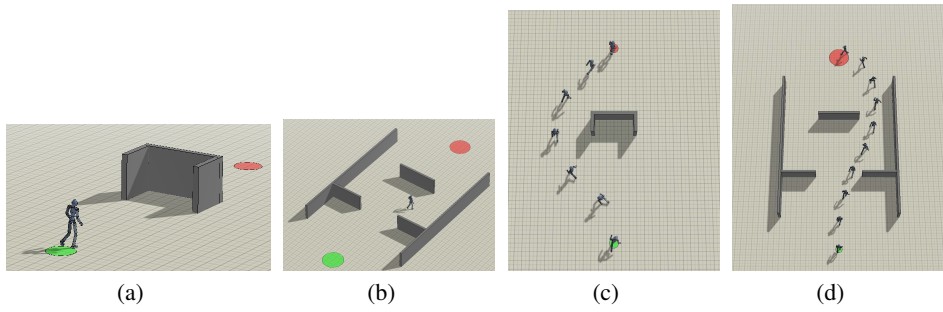

Figure 4: Cluttered environments for navigation tasks.

## 4.2 PLANNING AND CONTROL WITH LEARNED HIERARCHY

In the second experiment, we further demonstrate the capability of DISH framework to perform navigation tasks in cluttered environments (shown in Fig. 4). Since the humanoid character with the baseline models either kept falling or failed to walk in a desired direction, we omit the comparisons with the baselines in this task. The navigation reward is designed as a sum of two components: penalty for distance from the goal and penalty for collision with obstacles. As shown in Figs. 4(c) and 4(d) as well as in the supplementary video, the humanoid equipped with the DISH policy is able to not only escape a bug trap that cannot be overcome with greedy algorithms (i.e. without planning), but also navigate through obstacle regions successfully. Note that, unlike the HRL algorithms, the proposed hierarchical policy trained using the imitation tasks can be directly applied to the navigation tasks. It shows the generalization power of reasoning process; utilizing the internal model and the command-conditioned policy, the agent becomes able to plan and control its motions to adapt to new tasks and environments.

## 5 CONCLUSION

We proposed a framework to learn a hierarchical policy for an RL agent. In the proposed policy, the high-level loop plans the agent's motion by predicting its low-dimensional "task-specific" futures and the low-level loop maps the high-level commands into actions while actively reacting to the environment using its own state feedback loop. This sophisticated separation was able to emerge because two loops operated in different scales; the high-level planning loop only focuses on task-specific low-dimensional aspects in a coarser time scale, which enables it to plan relatively long-term futures. In order to learn the internal model for planning, we took advantage of recent advances in self-supervised learning of sequential data, while the low-level control policy is learned using a deep RL algorithm. By alternately optimizing both the LVM and the policy, the proposed framework was able to construct a meaningful internal model as well as a versatile control policy.

As future works, it would be interesting to incorporate visual inputs into the high-level reasoning module as suggested by Merel et al. (2019a). Though only continuous latent variables were considered in our framework, utilizing discrete variables such as a notion of *logics* or *modes* (Toussaint et al., 2018) also seems to be a promising direction. Lastly, besides imitation of experts, an agent should be able to learn from play data (Lynch et al., 2019) or from its own intrinsic motivation (Sharma et al., 2019).

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

# A  APPENDIX

## A.1  CONTROL-INFERENCE DUALITY

One theoretical concept this work extensively takes advantage of is the duality between optimal control (OC) and probabilistic inference (Levine, 2018; Todorov, 2008; Rawlik et al., 2012). The idea is that, if we consider an artificial binary observation whose emission probability is given by the exponential of a negative cost, an OC problem can be reformulated as an equivalent inference problem. In this case, the objective is to find the trajectory or control policy that maximizes the likelihood of the observations along the trajectory. One advantage of this perspective is that in order to solve the OC or RL problems, we can adopt any powerful and flexible inference methods, e.g., the expectation propagation (Toussaint, 2009), the particle filtering (Ha et al., 2018a; Piche et al., 2019), or the inference for Gaussian processes (Mukadam et al., 2018). In addition to utilizing efficient inference methods, this work also enjoys the duality to transform a multi-task RL problem into a generative model learning problem, which enables an agent to distill a low-dimensional representation and a versatile control policy in a combined framework.

## A.2  REINFORCEMENT LEARNING AS PROBABILISTIC INFERENCE

For easier reference, we restate the RL problem and the controlled trajectory distribution here:

$$\theta^* = \arg\max_{\theta} \mathbb{E}_{q_\theta(\mathbf{s}_{1:K}, \mathbf{a}_{1:K})} \left[ \sum_{k=1}^{K} \tilde{r}_k(\mathbf{s}_k, \mathbf{a}_k) \right], \tag{11}$$

$$q_\theta(\mathbf{s}_{1:K}, \mathbf{a}_{1:K}) \equiv p(\mathbf{s}_1) \prod_{k=1}^{K} p(\mathbf{s}_{k+1}|\mathbf{s}_k, \mathbf{a}_k) \pi_\theta(\mathbf{a}_k|\mathbf{s}_k), \tag{12}$$

respectively. It is well known in the literature that the above optimization (Equation 11) also can be viewed as a probabilistic inference problem for a certain type of graphical models (Levine, 2018; Todorov, 2008; Rawlik et al., 2012). Suppose we have an artificial binary random variable $o_t$, called the *optimality variable*, whose emission probability is given by exponential of a state-dependent reward, i.e.,

$$p(o_k = 1|\mathbf{s}_k) = \exp(r_k(\mathbf{s}_k)), \tag{13}$$

and the action prior $p(\mathbf{a}_k)$ defines the *uncontrolled* trajectory distribution (see also Fig. 2(a)):

$$p(\mathbf{s}_{1:K}, \mathbf{a}_{1:K}) \equiv p(\mathbf{s}_1) \prod_{k=1}^{K} p(\mathbf{s}_{k+1}|\mathbf{s}_k, \mathbf{a}_k) p(\mathbf{a}_k). \tag{14}$$

Then we can derive the evidence lower-bound (ELBO) for the variational inference:

$$\log p(O_{1:K}) = \log \int p(O_{1:K}|\mathbf{s}_{1:K}) p(\mathbf{s}_{1:K}, \mathbf{a}_{1:K}) d\mathbf{s}_{1:K} d\mathbf{a}_{1:K}$$

$$= \log \int p(O_{1:K}|\mathbf{s}_{1:K}) p(\mathbf{s}_{1:K}, \mathbf{a}_{1:K}) \frac{q_\theta(\mathbf{s}_{1:K}, \mathbf{a}_{1:K})}{q_\theta(\mathbf{s}_{1:K}, \mathbf{a}_{1:K})} d\mathbf{s}_{1:K} d\mathbf{a}_{1:K}$$

$$\geq \mathbb{E}_{q_\theta(\mathbf{s}_{1:K}, \mathbf{a}_{1:K})} \left[ \sum_{K=1}^{K} \left( \log p(O_k|\mathbf{s}_k) - \log \frac{\pi_\theta(\mathbf{a}_k|\mathbf{s}_k)}{p(\mathbf{a}_k)} \right) \right]$$

$$= \mathbb{E}_{q_\theta(\mathbf{s}_{1:K}, \mathbf{a}_{1:K})} \left[ \sum_{k=1}^{K} r_k(\mathbf{s}_k) - \log \frac{\pi_\theta(\mathbf{a}_k|\mathbf{s}_k)}{p(\mathbf{a}_k)} \right] \equiv \mathcal{L}(\theta). \tag{15}$$

The ELBO maximization in Equation 15 becomes equivalent to the reinforcement learning in Equation 11 by choosing an action prior $p(\mathbf{a}_k)$ and parameterized policy family $\pi_\theta(\mathbf{a}_k|\mathbf{s}_k)$ to match $\tilde{r}_k = r_k - \log \frac{\pi_\theta}{p}$[3]. Similar to Equation 19, the above maximization means to find the control policy $\pi_\theta$ resulting in the variational distribution that best approximates the posterior trajectory distribution when all the optimality variables were observed $p(\mathbf{s}_{1:K}, \mathbf{a}_{1:K}|O_{1:K} = 1)$.

---

[3]For example, when $p(\mathbf{a}_k)$ and $\pi_\theta(\mathbf{a}_k|\mathbf{s}_k)$ are given as Gaussian with the same covariance, $\log \frac{\pi_\theta}{p}$ encodes quadratic penalty on the control effort; when $p(\mathbf{a})$ is given as an uninformative uniform distribution, $\log \frac{\pi_\theta}{p}$ becomes the entropy regularization term in the maximum entropy reinforcement learning (Ziebart et al., 2008; Haarnoja et al., 2018b).

### A.3 Self-Supervised Learning of Latent Dynamical Models

Self-supervised learning is an essential approach that allows an agent to learn underlying dynamics from sequential high-dimensional sensory inputs. The learned dynamical model can be utilized to predict and plan the future state of the agent. By assuming that observations were emerged from the low-dimensional latent states, the learning problems are formulated as latent model learning, which includes an intractable posterior inference of latent states for given input data (Karl et al., 2017; Krishnan et al., 2017; Fraccaro et al., 2017; Ha et al., 2018b).

Suppose that a set of observation sequences $\{\mathbf{s}_{1:K}^{(n)}\}_{n=1,...,N}$ is given, where $\mathbf{s}_{1:K}^{(n)} \equiv \{\mathbf{s}_k; \forall k = 1, ..., K\}^{(n)}$ are i.i.d. sequences of observation that lie on (possibly high-dimensional) data space $\mathcal{S}$. The goal of the self-supervised learning problem of interest is to build a probabilistic model that well describes the given observations. The problem is formulated as a maximum likelihood estimation (MLE) problem by parameterizing a probabilistic model with $\phi$:

$$\phi^* = \arg\max_\phi \sum_n \log p_\phi(\mathbf{s}_{1:K}^{(n)}). \tag{16}$$

For latent dynamic models, we assume that the observations are emerged from a latent dynamical system, where a latent state trajectory, $\mathbf{z}_{1:K} \equiv \{\mathbf{z}_k; \forall k \in 1, ..., K\}$, lies on a (possibly low-dimensional) latent space $\mathcal{Z}$:

$$p_\phi(\mathbf{s}_{1:K}) = \int p_\phi(\mathbf{s}_{1:K}|\mathbf{z}_{1:K})p_\phi(\mathbf{z}_{1:K})d\mathbf{z}_{1:K}, \tag{17}$$

where $p_\phi(\mathbf{s}_{1:K}|\mathbf{z}_{1:K})$ and $p_\phi(\mathbf{z}_{1:K})$ are called a conditional likelihood and a prior distribution, respectively. Since the objective function (Equation 16) contains the intractable integration, it cannot be optimized directly. To circumvent the intractable inference, a variational distribution $q(\cdot)$ is introduced and then a surrogate loss function $\mathcal{L}(q, \phi; \mathbf{s}_{1:K})$, which is called the evidence lower bound (ELBO), can be considered alternatively:

$$\log p_\phi(\mathbf{s}_{1:K}) = \log \int p_\phi(\mathbf{s}_{1:K}|\mathbf{z}_{1:K})p_\phi(\mathbf{z}_{1:K})d\mathbf{z}_{1:K}$$
$$\geq \mathbb{E}_{q(\mathbf{z}_{1:K})} \left[\log \frac{p_\phi(\mathbf{s}_{1:K}|\mathbf{z}_{1:K})p_\phi(\mathbf{z}_{1:K})}{q(\mathbf{z}_{1:K})}\right]$$
$$\equiv \mathcal{L}(q, \phi; \mathbf{s}_{1:K}), \tag{18}$$

where $q(\cdot)$ can be any probabilistic distribution over $\mathcal{Z}$ of which support includes that of $p_\theta(\cdot)$. Note that the gap between the log-likelihood and the ELBO is the Kullback-Leibler (KL) divergence between $q(\mathbf{z})$ and the posterior $p_\theta(\mathbf{z}_{1:K}|\mathbf{s}_{1:K})$:

$$\log p_\phi(\mathbf{s}_{1:K}) - \mathcal{L}(q, \phi; \mathbf{s}_{1:K}) = D_{KL}(q(\mathbf{z}_{1:K})||p_\phi(\mathbf{z}_{1:K}|\mathbf{s}_{1:K})). \tag{19}$$

One of the most general approaches is the expectation-maximization (EM) style optimization where, alternately, (i) E-step denotes an inference procedure where an optimal variational distribution $q*$ is computed for given $\phi$ and (ii) M-step maximizes the ELBO w.r.t. model parameter $\phi$ for given $q*$.

Note that if we construct the whole inference and generative procedures as one computational graph, all the components can be learned by efficient end-to-end training (Karl et al., 2017; Krishnan et al., 2017; Fraccaro et al., 2017; Ha et al., 2018b). In p articular, Ha et al. (2018b) proposed the adaptive path-integral autoencoder (APIAE), a framework that utilizes the optimal control method; this framework is suitable to this work because we want to perform the planning in the learned latent space. APIAE considers the state-space model in which the latent states are governed by a stochastic dynamical model, i.e., the prior $p_\phi(\mathbf{z}_{1:K})$ is a probability measure of a following system:

$$\mathbf{z}_{k+1} = f_\phi(\mathbf{z}_k) + \sigma_\phi(\mathbf{z}_k)\mathbf{w}_k, \ \mathbf{z}_0 \sim p_0(\cdot), \ \mathbf{w}_k \sim \mathcal{N}(0, I). \tag{20}$$

Additionally, a conditional likelihood of sequential observations is factorized along the time axis:

$$p_\phi(\mathbf{s}_{1:K}|\mathbf{z}_{1:K}) = \prod_{k=1}^K p_\phi(\mathbf{s}_k|\mathbf{z}_k). \tag{21}$$

If the variational distribution is parameterized by the control input $\mathbf{u}_{1:K-1}$ and the initial state distribution $q_0$ as:

$$\mathbf{z}_{k+1} = f_\phi(\mathbf{z}_k) + \sigma_\phi(\mathbf{z}_k)\left(\mathbf{u}_k + \mathbf{w}_k\right), \; \mathbf{z}_0 \sim q_0(\cdot), \; \mathbf{w}_k \sim \mathcal{N}(0, I), \tag{22}$$

the ELBO can be written in the following form:

$$\mathcal{L} = \mathbb{E}_{q_\mathbf{u}}\left[ \log p_\phi(\mathbf{s}_{1:K}|\mathbf{z}_{1:K}) + \log \frac{p_0(\mathbf{z}_0)}{q_0(\mathbf{z}_0)} - \sum_{k=1}^{K-1} \frac{1}{2}\|\mathbf{u}_k\|^2 + \mathbf{u}_k^\top \mathbf{w}_k \right]. \tag{23}$$

Then, the problem of finding the optimal variational parameters $\mathbf{u}^*$ and $q_0^*$ (or equivalently, the best approximate posterior) can be formulated as a stochastic optimal control (SOC) problem:

$$\mathbf{u}^*, \; q_0^* = \underset{\mathbf{u}, q_0}{\arg\min} \; \mathbb{E}_{q_\mathbf{u}(\mathbf{z}_{1:K})}\left[ V(\mathbf{z}_{1:K}) + \sum_{k=1}^{K-1} \frac{1}{2}\|\mathbf{u}_k\|^2 + \mathbf{u}_k^\top \mathbf{w}_k \right], \tag{24}$$

where $V(\mathbf{z}_{1:K}) \equiv -\log \frac{p_0(\mathbf{z}(0))}{q_0(\mathbf{z}(0))} - \sum_{k=1}^{K} \log p_\phi(\mathbf{s}_k|\mathbf{z}_k)$ serves as a state cost of the SOC problem. Ha et al. (2018b) constructed the differentiable computational graph that resembles the path-integral control procedure to solve the above SOC problem, and trained the whole architecture including the latent dynamics, $p_0(\mathbf{z})$, $f_\phi(\mathbf{z})$ and $\sigma_\phi(\mathbf{z})$, and the generative network, $p_\phi(\mathbf{s}|\mathbf{z})$ through the end-to-end training.

### A.4 Planning by Particle Filtering

---

**Algorithm 2** Particle Filtering with Internal Model for Planning

---

1: Initialize $\forall i \in \{1, ..., N_{\text{particle}}\} : \mathbf{z}_1^{(i)} \sim q_\phi(\cdot|\mathbf{s}_{:\text{cur}})$ and $w_1^{(i)} = 1/N_{\text{particle}}$
2: **for** $k = 2, ..., K_{\text{plan}}$ **do**
3:     **for** $i = 1, ..., N_{\text{particle}}$ **do**
4:         $\mathbf{z}_k^{(i)} = f_\phi(\mathbf{z}_{k-1}^{(i)}) + \sigma_\phi(\mathbf{z}_{k-1}^{(i)})\left(\mathbf{h}_{k-1}^{(i)} + \mathbf{w}_{k-1}^{(i)}\right), \; \mathbf{w}_{k-1}^{(i)} \sim \mathcal{N}(0, I)$
5:         $\mathbf{s}_k^{(i)} \sim \mathcal{N}\left(\mu_\phi(\mathbf{z}_k^{(i)}), \Sigma_\phi(\mathbf{z}_k^{(i)})\right).$
6:         $w_k^{(i)} = w_{k-1}^{(i)} \exp(r_k(\mathbf{s}_k^{(i)}))$
7:     **end for**
8:     $w_k^{(i)} = w_k^{(i)} / \sum_j w_k^{(j)}, \; \forall i \in \{1, ..., N_{\text{particle}}\}$
9:     Resample $\{\mathbf{z}_{1:k}^{(i)}, \mathbf{w}_{1:k}^{(i)}\}$ if $\left(\sum_i (w_k^{(i)})^2\right)^{-1} < N_{\text{particle}}/3$
10: **end for**
11: **return** $\mathbf{h}_1^* = \sum_i w_{K_{\text{plan}}}^{(i)} \mathbf{w}_1^{(i)}$     $\triangleright \; \mathbf{h}_k^* = \sum_i w_{K_{\text{plan}}}^{(i)} \mathbf{w}_k^{(i)}, \forall k = 1, .., K_{\text{MPC}}$ for general MPC cases

---

At each time step $\delta t$, the high-level planner takes the current state as an argument and required to output the commands by predicting the future trajectory and corresponding reward $r_k(\cdot)$. We adopted the particle filter algorithm to perform such the reasoning and the pseudo code is shown in Algorithm 2. The particle filter algorithm attempts to represent the posterior distribution using a set of samples. The algorithm first samples $N_{\text{particle}}$ initial latent states using the inference network (which is a part of the learned internal model) and assigns the same weights for them. During the forward recursion, the particles are propagated using the latent dynamics of the internal model (line 4), and the corresponding configurations are generated through the learned model (line 5). The weights of all particles are then updated based on the reward of the generated configurations (line 6 and 8); i.e., the particles that induce higher reward values will get higher weights. If only a few samples have weights effectively, i.e., if the weights collapse, the algorithm resamples the particles from the current approximate posterior distribution to maintain the effective sample size (line 9). After the forward recursion over the planning horizon, the optimal commands are computed as a linear combination of the initial disturbances; i.e., it is given by the expected disturbance under the posterior transition dynamics (Kappen & Ruiz, 2016).

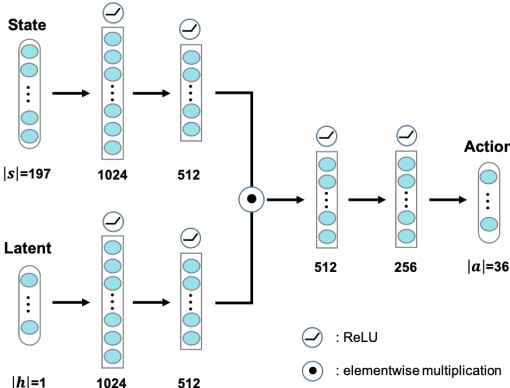

Figure 5: The network architecture of low-level policy network.

## A.5 TRAINING LOW-LEVEL POLICY

For the training algorithm for low-level policy network ($\pi_\theta$), we extend motion imitation approach (Peng et al., 2018) to multi-task scheme; we construct value networks parameterized by neural network with size $[197, 1024, 512, 1]$ for each task (three in our experiments), and the low-level policy network (actor network) taking a state feature $\mathbf{s}$ and a latent variable $\mathbf{h}$ as inputs to determine an action $\mathbf{a}$ as illustrated in Fig. 5. The imitation reward is given as following:

$$r_t = 0.3 r_t^{\text{root}} + 0.2 r_t^{\text{pose}} + 0.15 r_t^{\text{vel}} + 0.15 r_t^{\text{ee}} + 0.2 r_t^{\text{com}} \tag{25}$$

$$r_t^{\text{root}} = \exp\left(-0.5||\hat{\mathbf{p}}_t^r - \mathbf{p}_t^r||_2^2 - 0.5||\hat{\dot{\mathbf{p}}}_t^r - \dot{\mathbf{p}}_t^r||_2^2 - 0.5||\hat{\mathbf{q}}_t^r - \mathbf{q}_t^r||_2^2 - 0.05||\hat{\dot{\mathbf{q}}}_t^r - \dot{\mathbf{q}}_t^r||_2^2\right)$$

$$r_t^{\text{pose}} = \exp\left(-2\sum||\hat{\mathbf{q}}_t^j - \mathbf{q}_t^j||_2^2\right) \qquad r_t^{\text{vel}} = \exp\left(-0.1\sum||\hat{\dot{\mathbf{q}}}_t^j - \dot{\mathbf{q}}_t^j||_2^2\right)$$

$$r_t^{\text{ee}} = \exp\left(-40\sum||\hat{\mathbf{p}}_t^e - \mathbf{p}_t^e||_2^2\right) \qquad r_t^{\text{com}} = \exp\left(-||\hat{\dot{\mathbf{p}}}_t^c - \dot{\mathbf{p}}_t^c||_2^2\right)$$

where $\mathbf{q}_t$ and $\mathbf{p}_t$ represent angle and global position while $\hat{\ }$ represent those of the reference.[4] As reference motion data, three motion capture clips, turning left ($\mathbf{t} = [1,0,0]$), going straight ($\mathbf{t} = [0,1,0]$), turning right ($\mathbf{t} = [0,0,1]$) from the Carnegie Mellon University motion capture (CMU mocap) database are utilized. Following the reference, PPO with same hyperparameters is used for RL algorithm. Since the internal model does not exist at the first iteration ($l = 1$), the high-level planner is initialized by $q_\phi(\mathbf{h}|\mathbf{s}; t) = \mathbf{w}^T \mathbf{t}$ where $\mathbf{w} = [-1, 0, 1]$. After the first iteration, high-level planner computes a command $\mathbf{h}_t$ that makes the model to best follow the horizontal position of the reference motion for 5 seconds ($\delta t = 0.1s$ and $K_{plan} = 50$). The corresponding reward function is as following:

$$r_k = -||\hat{\mathbf{p}}_{h,k}^r - \mathbf{p}_{h,k}^r||_2^2 \tag{26}$$

where $\mathbf{p}_{h,k}$ is the horizontal components of position vector at time step $k$.

## A.6 TRAINING INTERNAL MODELS

Internal models of DISH is trained to maximize the ELBO in Equation 23 by using the APIAE approach (Ha et al., 2018b) with hyperparameters as following: 3 adaptations ($R = 4$), 10 time steps ($K = 10$), 32 samples ($L = 32$), and time interval of 0.1s ($\delta t = 0.1$). The network architectures of transition network and inference network are shown in Fig 6.

For the baselines, the transition functions, $f_\phi(\mathbf{x}_{k+1}|\mathbf{x}_k, \mathbf{y}_k)$, were parameterized by neural networks having same architectures as DISH except for the input variables as shown in Table 3. The loss

---

[4]Each superscript denotes as following: $r$: root (pelvis), $j$: local joints, $e$: end-effectors (hands and feet), $c$: center-of-mass.

Table 3: Input variables of baseline transition models

| input variables | DISH | $sas'$ | $shs'$ | $zaz'$ |
|---|---|---|---|---|
| $\mathbf{x}_k$ | $\mathbf{z}_k$ | $\mathbf{s}_k$ | $\mathbf{s}_k$ | $\mathbf{z}_k$ |
| $\mathbf{y}_k$ | $\mathbf{h}_k$ | $\mathbf{a}_k$ | $\mathbf{h}_k$ | $\mathbf{a}_k$ |

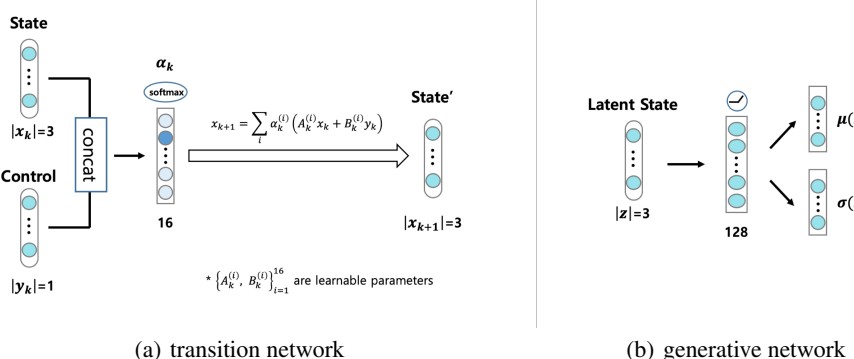

(a) transition network          (b) generative network

Figure 6: The network architecture of internal model.

function for baseline is as following:

$$\mathcal{L}_{sas'}(\phi) = \sum_{k=1}^{K} ||\mathbf{s}_k - \tilde{\mathbf{s}}_k||_2^2, \quad \tilde{\mathbf{s}}_k = f_\phi^{sas'}(\tilde{\mathbf{s}}_{k-1}, \mathbf{a}_{k-1}), \tag{27}$$

$$\mathcal{L}_{shs'}(\phi) = \sum_{k=1}^{K} ||\mathbf{s}_k - \tilde{\mathbf{s}}_k||_2^2, \quad \tilde{\mathbf{s}}_k = f_\phi^{shs'}(\tilde{\mathbf{s}}_{k-1}, \mathbf{h}_{k-1}), \tag{28}$$

$$\mathcal{L}_{zaz'}(\phi) = \sum_{k=1}^{K} ||\mathbf{s}_k - g(\tilde{\mathbf{z}}_k)||_2^2, \quad \tilde{\mathbf{z}}_k = f_\phi^{zaz'}(\tilde{\mathbf{z}}_{k-1}, \mathbf{a}_{k-1}), \tag{29}$$

where $\tilde{\mathbf{s}}_1 = \mathbf{s}_1$, $\tilde{\mathbf{z}}_1$ is latent state for $\mathbf{s}_1$ inferred by VAE, and $g(\cdot)$ is learned generative network of VAE.

## A.7 TASK CONFIGURATIONS

**Trajectory Following Tasks:** Planning reward $r_t$ penalizes the distance between the horizontal position of the root of humanoid character $\mathbf{p}_k^r$ and the that of reference trajectory $\bar{\mathbf{p}}_k$:

$$r_k = -||\bar{\mathbf{p}}_k - \mathbf{p}_{h,k}^r||_2^2. \tag{30}$$

**Navigation Tasks:** Planning cost $r_t$ penalizes the distance between the horizontal position of the root of humanoid character $\mathbf{p}_k^r$ and the that of the goal $\mathbf{p}_{\text{goal}}$ and the collision with obstacles, while giving a reward on arrival:

$$r_k = -||\mathbf{p}_{\text{goal}} - \mathbf{p}_{h,k}^r||_2^2 - 10^5 \times (\text{IS\_CRASHED}) + 10^4 \times (\text{IS\_REACHED}). \tag{31}$$

## A.8 FURTHER RESULTS

Table 4 reports the RMS between reference, planned, and executed trajectories for each tasks. As illustrated in the table, DISHs showed the best performance. Although $shs'$ sometimes showed smaller errors for the difference between the planed and reference trajectories, the errors between the reference and executed trajectory of DISHs are always smallest. This demonstrates that DISHs best learn the internal dynamics of the humanoid, making the most valid predictions for future motion.

Table 4: Comparison of RMS errors between reference, planned, and executed trajectories for different types of internal models.

| | task1 (turn left) | | | task2 (go straight) | | | task3 (turn right) | | |
|---|---|---|---|---|---|---|---|---|---|
| | ‖ref-true‖ | ‖plan-ref‖ | ‖plan-true‖ | ‖ref-true‖ | ‖plan-ref‖ | ‖plan-true‖ | ‖ref-true‖ | ‖plan-ref‖ | ‖plan-true‖ |
| DISH | **0.1290** | **0.1364** | **0.0743** | 0.2073 | 0.1705 | **0.1073** | **0.1550** | **0.1661** | **0.0974** |
| DISH+ | 0.1466 | 0.1704 | 0.1223 | **0.1177** | 0.1075 | 0.1177 | 0.1711 | 0.1747 | 0.0988 |
| $sas'$ | F | 0.2474 | F | F | 0.0660 | F | F | 0.2178 | F |
| $shs'$ | 0.2525 | 0.2036 | 0.3167 | 0.1385 | **0.0561** | 0.1280 | 0.2767 | 0.2140 | 0.2247 |
| $zaz'$ | F | 0.2731 | F | F | 0.1994 | F | F | 0.3044 | F |

Comparing DISH ($L = 1$) and DISH+ ($L = 2$), we can observe that DISH outperforms in the turning tasks while showing the worse performance in going straight. This is because the high-level planner is initialized to output only one of $\{-1, 0, 1\}$ (as shown in Appendix A.5), so the corresponding low-level policy of DISH is trained only around $\mathbf{h} \in \{-1, 0, 1\}$ rather than along the continuous $\mathbf{h}$ values. As a result, the DISH agent is only good at radical turns (not smooth turns), making it difficult to stabilize its heading direction properly. The ability to turn smoothly is obtained in the next iteration where the proper reasoning module is equipped, thus, although it lost some ability to turn fast, the DISH+ agent achieves the better ability to walk straight and the increased average performance (see Table 2).

Fig. 7 shows rollout samples by varying the control values. Except for DISHs, the generated trajectories are very noisy, which indicates that the baseline internal models are not suitable for planning the future movements of the humanoid.

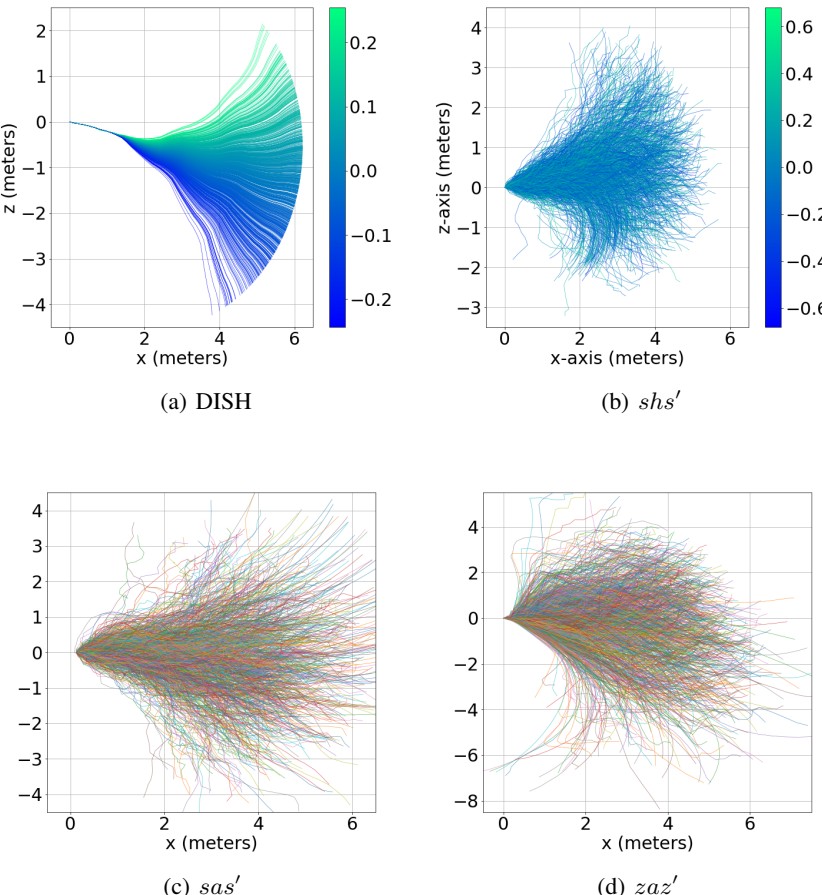

(a) DISH

(b) $shs'$

(c) $sas'$

(d) $zaz'$

Figure 7: Rollout samples from different types of internal models. (a) and (b) is colored by latent control value.

