# OpenReview forum: "Learning to Reason: Distilling Hierarchy via Self-Supervision and Reinforcement Learning"
_ICLR.cc/2020/Conference — Reject_

### Official Review · AnonReviewer2 · 2019-10-18
**Official Blind Review #2**

**Rating:** 6

**Review:**

*Edit: original score was a weak reject (3), updating to a weak accept (6) in light of revisions.*

This work implements a hierarchical control scheme for a high-dimensional control problem (locomotion using a humanoid body).  The hierarchy consists of a high-level module that plans in an abstract space of "intention", and the intention variables serve as inputs, along with state, to a low-level controller that actually executes the movements.  The premise is that a lower-level controller should be usable for multiple tasks, and should be able to be commanded by a lower-dimensional intention input.  I find the basic ideas presented clear, the literature reviewed reasonably well, and the motivation and setting to be very interesting.  The video summary is valuable.

My main concerns have to do with presentation, but I think they are relatively significant concerns.  As the draft currently stands, I would, somewhat regrettably, be inclined to reject the submission (marginally).  I think revisions could seriously improve this paper and incline me towards acceptance.

Algorithm 1 indicates that the learning of the low-level controller will be done jointly with the learning of the latent model and planning using the high-level, learned intention space.  In the experiment, it is indicated that the low-level controller is pretrained.  This points to a couple issues that are not clear in the draft:
(1) Presumably this pretraining is necessary and things do not work without it.  Indeed, it is hard to imagine that the movements will be well grounded to human motion capture movements without this pretraining.  Does the algorithm work as written or is pretraining a fundamentally essential step?  There are no settings, even toy settings, where the algorithm as written is shown to be effective.
(2) The authors should be clearer how they conduct the pretraining which involves learning the low-level controller.
(3) I'm not clear how updating the low-level controller is effective in the algorithm.  While I understand why it makes sense to plan in the intention space of the pre-trained controller, and I understand why learning a model is a core part of planning, it would seem like fine-tuning the low-level controller could make the movements deviate considerably from the initial movement space and maybe even eliminate the ability of the low-level policy to express movements that are not used early in training. So essentially, while the planning in the low-D space makes sense and the learning of the model makes sense, the low-level controller update seems possibly to not make sense, and there aren't experiments showing that step helps.

Is it just a coincidence that the intention space (h) is one-of-three and the low-d state space (z) is 3-dimensional as well?  Or are these both selected with sort of going straight vs turning left or right in mind?

The experiment section is generally very unclear, though details are made a little clearer from the video.  In the paper, there are a few points that need to be clearer:
(1) "ref", "plan", and  "true" are not well defined and it is unclear what these distances in Table 1 refer to precisely.  Clearly introduce what each of these refers to.  The authors simply say that there are imitation tasks but do not walk through what these terms refer to.
(2) In 4.1, the different structures are not adequately introduced.  The pointers to the figure 2 diagrams are essential, but there is no pointer for zaz', the pointers are only in the table (not in the text), there are grammar issues in the text and the text could be verbally clearer about the variants.
(3) shs' setting is a bit unclear. Basically, clarify briefly how planning is performed in this case.   Is a forward model still trained, but the model opperates with the full state space?  If so, presumably the forward model is much worse and then the planning approach is correspondingly bad, hence the poor rollouts?
(4) 4.2 is I think obviously inadequately described in the text and I can only assume was the result of rushing for the deadline?  The second experiment is essentially not presented in the text all aside from a still image.
(5) And for all of the experiments that rely on planning with a particle filter, details such as how reliable the filter is in generating useful control, how many samples are required, and possibly elements of compute speed would make much clearer how well the approach actually works.  Does the choice of planner matter at all?  A common, albeit relatively weak, baseline planning approach is CEM...would CEM work here?  I'd like to understand if the choice of particle filter is the author's default choice, which is fine if so, or if there is a positive assertion being made that the particle filter is particularly valuable.

I hope the authors will generally improve the exposition in the experiment section (4) during the revisions.

Overall, I find the paper well motivated in framing the problem (i.e. using model-based approaches to control the latent space of a low-level controller).  I also appreciate the scale of the problem (humanoid control is challenging, so this is not a toy problem).  I find the results a bit unclear, perhaps due to hurriedness in writing, so I find them a bit difficult to fully appreciate.  Nevertheless, the core contribution that I take away from this work is that there is a value to learning the low-dimensional state representation (z, via the LVM), relative to planning using a forward model on the full state (?...I'm still unclear on the presentation of this result, due to unclear exposition). Slightly more broadly, this is a good demonstration of using a planner jointly with a learned high-level command/intention representation, for a high-dimensional problem.

If I've understood this correctly, I'd be reasonably interested in this result. If the authors can both clarify the core results and communicate that the choices made in the algorithm are well thought through, I would be happy to adjust my score.


Relatively minor:

Abstract says 90-dimensional humanoid system, but later it is stated "34 degrees of freedom,
197 state features, and 36 action parameters". Where the 90 dimensions comes from is unclear.  Often people refer to number of actuators or DoFs.  Please adjust this or be more explicit.

In equation 9, f() is not very clearly specified. Is f() a nonlinear function (e.g. a neural network) or is it a linear function? It seems like it might as well be a linear function, since the authors propose to learn a latent dynamics model that is nonlinearly related to the state.

Typos in Fig 3 caption.


**Experience Assessment:**

I have published in this field for several years.

**Review Assessment: Checking Correctness Of Derivations And Theory:**

N/A

**Review Assessment: Checking Correctness Of Experiments:**

I assessed the sensibility of the experiments.

**Review Assessment: Thoroughness In Paper Reading:**

I read the paper thoroughly.

---

> ### Author Response · Authors · 2019-11-11
> **Initial response to reviewer2 (2/2)**
>
> Re: about the experiment section
> (1) We’ve clearly stated what “ref”, “plan”, and “true” mean in the revised paper.
> (2) We’ve added much more detailed descriptions of the different structures in 4.1 as well as in the appendix. There is actually no depicted graphical model for $zaz’$ so we stated the additional explanation for this model (as LVM counterpart of Fig. 2(a)).
> (3) Yes, your understanding is correct. After the internal model trained (which operates with the full state space), the reasoning is performed there. Since the dynamics should be learned in the full state space, it fails to capture the valid stochastic transition according to $\mathbf{h}$, thereby making the planning become worse. We’ve added the statements on this as well.
> (4) We’ve included the detailed experimental setting for 4.2, such as planning horizon and frequency, used reward functions, and the implication of the results.
> (5) For the internal model learning, we used 32 samples. It works with so few samples because the learning should be parallelized, but the minibatch training seems to alleviate the problem from the small sample size. We used 1024 samples for the planning and observed that larger sample sizes result in better planning results. A particle filter is just one instance of a reasoning module; if it iterates the whole procedure multiple times to compute a better command, then the planning algorithm becomes the adaptive path integral method. And if the resampling procedure is eliminated, it is equivalent to the widely-used CEM. CEM would work here as well here. We chose a particle filter because we thought its resampling procedure could be very useful for our task; during forward recursion, the procedure greedily gets rid of candidate motions in the wrong direction for imitation tasks and motions colliding with an obstacle for navigations.
>
> Re: minor comments
> - We explicitly state the dimensionality of data, as 197-dimensional state feature and 36-dimensional action space. The 90-dimensional state was just another form of data representation (position/orientation of each link vs. quaternion representation of each joint), and we omitted it for clarity.
> - f() is a nonlinear function which is parameterized as a neural network.
>
> ==============================================================================================================================
> Thank you again for your constructive comments. All of them were helpful to revise our manuscript. We would be happy to discuss further if you have any more comments. Please adjust your score if you think the revision has been done properly.

---

> ### Author Response · Authors · 2019-11-11
> **Initial response to reviewer2 (1/2)**
>
> Thank you for your kind review and suggestions. We admit that many parts of our draft were incomplete (due to hurriedness in writing…), but we’ve made a big revision of the manuscript especially in the main algorithm and experiment sections and hope that the paper has been improved a lot particularly in terms of presentation.
>
> ==============================================================================================================================
> Re: Algorithm 1 and pretraining
> - We realized that the draft didn’t properly describe the overall learning procedure. The main learning algorithm runs in the same way from the first iteration, but because the agent doesn’t have an internal model in the first iteration, we manually set the high-level reasoning to output the predefined values, $\{-1, 0, 1\}$, for each task and we somewhat interpreted this first iteration as a pretraining.
> - Identifying the low-dimensional internal model with imitation learning tasks is considered just to boost the learning procedure (and also because the Mocap data as well as a suitable imitation learning algorithm, DeepMimic, is available. DeepMimic is almost the only algorithm making humanoid walk like a human, in my personal opinion.). Otherwise, we might have to design a proper curriculum of tasks in different outer-loop iterations from crawling, sitting, standing to walking, turning, running, just like how we learned to walk. Instead of a curriculum, an RL objective function for intrinsic motivation also can be introduced to acquire skills.
> - In the $l$th iteration, the reasoning module might output commands that the low-level policy hasn’t yet provided in the previous iteration. This is because interpolation for the low-dim internal model makes sense much more than that for the high-dim control policy or simply because a new set of tasks can be provided. The low-level policy then needs to be trained further to execute the new commands. To show that, we added the comparisons between DISH (L=1) and DISH+ (L=2); in our example of locomotion, while good at rapid turn motions, the DISH agent can’t turn smoothly (e.g., for $\mathbf{h}=0.5$) because the low-level policy hasn’t been fed such commands, but DISH+ learns those smooth turns.
>
> Re: about the dimensionality of the latent space
> Yes, it is actually a coincidence that the (initial) intentions are one-of-three and the latent state is the 3-dimensional. We empirically found that the latent dynamics is encoded best in the 3-dimensional space. We believe that, ultimately, the dimensionality of the latent space also should be learned but the current gradient-based learning algorithms seem not very suitable for doing so.

---

> ### Comment · AnonReviewer2 · 2019-11-11
> **Authors meaningfully improved draft + score update.**
>
> As of the revision visible at the present moment, I remain not entirely happy with the presentation of the experiments, but these concerns are now essentially just issues related to what I believe to be clear writing.  I encourage a few more read-throughs by the authors with an eye towards editing for clarity.  In particular, I still think the authors don't clearly define their setting succinctly at the beginning of the "Experiment" section.  And in the subsections of that section, they should open by very briefly defining the tasks.  The appendices provide additional information, but they should be more explicitly referenced, with an indication of what they contain.
>
> The above comments notwithdstanding, the authors have made meaningful revisions, and the content required to fully interpret the results is now present.  I think this work is both interesting and contains valuable contributions.  So I will update my score to a weak accept (6).  Ultimately, the novel contributions of this work involve the ability to perform planning using the low-dimensional space to reuse the humanoid motor skills, and these contributions warrant acceptance to the conference.  This is a challenging problem, so this work does provide a somewhat satisfying demonstration.  The ablation experiments make the work valuable to build upon.

---

> > ### Author Response · Authors · 2019-11-14
> > **Further revisions in experiment section**
> >
> > Thank you for your positive comments and suggestions. We've revised the experiment section to deliver our setting and results more clearly.
> >
> > - We explain the model, simulation environment, learning algorithms, and hyper-parameters with explicit references to the appendices at the beginning of the section.
> > - The subsections are then started with the task specifications and the settings for the baseline algorithms, with the proper references to appendices.
> >
> > We hope the experiment section has now become much clearer and we will continue to revise the entire manuscript.

---

### Official Review · AnonReviewer3 · 2019-10-21
**Official Blind Review #3**

**Rating:** 1

**Review:**

This paper presents a framework for learning hierarchical policies using a latent variable conditioned policy operating at the low level, with model based planning at the high level. Unlike prior work which does hierarchical reinforcement learning, the key technical contribution of this work is that they use planning with a latent dynamics model as their high level policy. They demonstrate the method on a humanoid walking task in the DeepMimic [1] environment.

While the idea is well motivated, this paper should be rejected, primarily due to a lack of experimental results. In particular, the experiments (1) are missing several critical details about the experimental setup, (2) the experimental setup differs significantly from the claims of self-supervision and multi-task RL made in the introduction/method, and (3) there is no comparison to any prior work. Without these, it is impossible to determine if any of the claims made about the proposed method are empirically true.

First, no information is provided about the reward function used, the horizon of the tasks, or about the planning parameters or policy learning parameters. As currently stated, I don't think any of the results in the paper could be reproduced. Furthermore, the reward function and task horizon used are necessary to determine the difficulty of the proposed tasks.

Second, the title and method section would imply that the method is self-supervised, specifically in how the latent dynamics model is learned. While the samples used to train the latent dynamics model are taken from the agent's experience, the latent conditioned policy is trained (1) with ground truth task reward, and (2) is actually pre-trained on demonstrations of the tasks with ground truth skill labels. This suggests that much of the actual skill learning is done offline in this pre-training stage - with full supervision. As a result, this would make the learning of the latent dynamics model much easier, since the sub-policies have already converged to different behaviors for different values of h. Without this pre-training, learning the low level policies and the LVM jointly would be much more challenging. Hence it seems that the "self-supervised" learning of the LVM is actually heavily dependent on the full supervision used in the pre-training stage. Additionally, the demonstrations used for the pre-training correspond to the same 3 tasks that the agent is later evaluated on (moving forward, left, right). So the method receives full supervision on the test tasks, so the experiments do not actually reflect generalization in multi-task RL as claimed.

Lastly, and most importantly, there are no comparisons to prior work. The only result shown is the trajectory error against 3 ablations of the proposed method. The reported numbers are error between the reference trajectory and ground truth, predicted plan and reference, and predicted plan and ground truth. First, it seems like the most important number here is the difference between the predicted plan and ground truth, for which numbers are missing for 2/3 ablations. Why is task success or reward not the reported number, and why is performance for 2/3 ablations missing? Additionally, there should be comparisons to existing work both in terms of hierarchical model free RL (for example Nachum et al [2]) and model based RL with latent dynamics models (Hafner et al [3]). The results as presented do not actually support that the proposed method performs better than existing work. The final result of the video of the agent doing a long horizon task also has no quantitative numbers, so again does not support that the proposed method is better.

Some other less significant points:
- there are typos throughout (for example "Figure 3: (a) Leanred latent model.").
- the tables and figures have very limited or no captions.
- The method section is difficult to follow and could use some figures which demonstrate the key technical contribution.
- Also from the method section it seems like the novelty is combining the latent conditioned policy learning from Haarnoja et al [4] and the latent dynamics learning/planning from Ha et al [5]. Is there an additional technical contribution beyond combining these two existing works? If so the method section should more clearly show it.
- There is a recent work (Sharma et al [6]), which also learns skill conditioned low level policies, and does model based planning in the space of skills to reach previously unseen goals. In this work the skill discovery is also totally unsupervised. The authors should add citation to this paper and clarify the differences between their work and this work.

[1] Xue Bin Peng, Pieter Abbeel, Sergey Levine, and Michiel van de Panne. DeepMimic: Example guided deep reinforcement learning of physics-based character skills.
[2] Ofir Nachum, Shane Gu, Honglak Lee, and Sergey Levine. Data-efficient hierarchical reinforcement learning
[3] Hafner, D., Lillicrap, T. P., Fischer, I., Villegas, R., Ha, D., Lee, H., and Davidson, J. Learning latent dynamics for planning from pixels
[4] Tuomas Haarnoja, Kristian Hartikainen, Pieter Abbeel, and Sergey Levine. Latent space policies for hierarchical reinforcement learning.
[5] Jung-Su Ha, Young-Jin Park, Hyeok-Joo Chae, Soon-Seo Park, and Han-Lim Choi. Adaptive path integral autoencoders: Representation learning and planning for dynamical systems.
[6] Archit Sharma, Shixiang Gu, Sergey Levine, Vikash Kumar, and Karol Hausman. Dynamicsaware unsupervised discovery of skills.



**Experience Assessment:**

I have published one or two papers in this area.

**Review Assessment: Checking Correctness Of Derivations And Theory:**

I assessed the sensibility of the derivations and theory.

**Review Assessment: Checking Correctness Of Experiments:**

I carefully checked the experiments.

**Review Assessment: Thoroughness In Paper Reading:**

I read the paper thoroughly.

---

> ### Author Response · Authors · 2019-11-11
> **Initial response to reviewer3 (2/2)**
>
> RE: about comparisons
> - Our initial descriptions about the experiment were insufficient and now we’ve updated them properly. For the ablation study, some numbers for $sas’$ and $zaz’$ are missing because these models failed to let the humanoid walk at all. For $sas’$ and $zaz’$, the reasoning modules should search over the 36-dimensional action space with the limited number of particles (1024 in our case). It is infeasible due to the curse of dimensionality and, given the fact that the humanoid has very unstable dynamics, the inaccurately planed action can’t make the humanoid walk.
> - In this work, we propose a hierarchical policy that, in a high-level, performs planning using a learned internal model. Section 4.2 demonstrates that the proposed policy learned via imitations directly can be applied to unseen navigation tasks, showing powerful generalization ability of reasoning/planning process. We don’t want to argue that our framework outperforms existing model-free HRL algorithms; even on top of the HRL algorithms, a reasoning module can be introduced (e.g., having more than two levels). What we try to argue in this paper is that, in order to construct such a reasoning module, low-dimensional latent state $\mathbf{z}$ as well as low-dimensional command $\mathbf{h}$ should be considered together. While most model-based RL methods with planning module in a higher-level utilize only one of $\mathbf{z}$ and $\mathbf{h}$, our ablation study shows that the scalable reasoning can be performed only through having both $\mathbf{z}$ and $\mathbf{h}$. We believe that this is a valid and meaningful argument, given the fact that high-dimensional motion planning and control is a longstanding challenging problem in robotics literature [6,7].
> - We will add the quantitative numbers/analysis for Section 4.2 as well.
>
> RE: relatively minor comments
> - We’ve addressed all the other comments, thanks. (fixed typos, strengthened captions, emphasized our contribution, cited and discussed the existing work)
>
> ==============================================================================================================================
> We hope our revision is valid enough so that you can reconsider the score of this submission. We would be happy to have further discussions as well.
>
>
> [1] Karl, Maximilian, et al. "Deep variational bayes filters: Unsupervised learning of state space models from raw data."
> [2] Krishnan, Rahul G., Uri Shalit, and David Sontag. "Structured inference networks for nonlinear state space models."
> [3] Fraccaro, Marco, et al. "A disentangled recognition and nonlinear dynamics model for unsupervised learning."
> [4] Ha, Jung-Su, et al. "Adaptive Path-Integral Autoencoders: Representation Learning and Planning for Dynamical Systems."
> [5] Sharma, Archit, et al. "Dynamics-aware unsupervised discovery of skills."
> [6] Vernaza, Paul, and Daniel D. Lee. "Learning and exploiting low-dimensional structure for efficient holonomic motion planning in high-dimensional spaces."
> [7] Ha, Jung-Su, Hyeok-Joo Chae, and Han-Lim Choi. "Approximate inference-based motion planning by learning and exploiting low-dimensional latent variable models."

---

> ### Author Response · Authors · 2019-11-11
> **Initial response to reviewer3 (1/2)**
>
> Thank you for your critical and constructive comments.  As you pointed out, our draft was missing several important details about the experimental setup, descriptions, and discussions, so we’ve extensively revised the manuscript to strengthen them. Followings are our response to your comments:
>
> Re: about the experimental details
> We’ve included all the experiment information about the reward functions, the policy and internal model structures, planning horizon and frequency, and so on. For the reproducibility, we will make the source code available in the final manuscript.
>
> Re: about the overall learning algorithm especially regarding self-supervised learning
> - I think our ideas on how to utilize self-supervised learning are somewhat different from yours. By self-supervised learning, we mean that the agent acquires the low-dimensional representations of observation sequences and its stochastic dynamics only to maximize the data likelihood (e.g., Eq.(7)) without any supervision about any forms of latent space [1,2,3,4].
> - Imitation learning tasks are considered just to boost the procedure of identifying the low-dimensional internal model (and also because the Mocap data as well as a suitable imitation learning algorithm, DeepMimic, is available. DeepMimic is almost the only algorithm making humanoid walk like a human, in my personal opinion.).  In the first main iteration where the agent doesn’t have any internal models, the low-level policy learns locomotions via the imitation learning with the manually set reasoning module and the internal model is trained via self-supervised learning.
> - Besides imitation, we might be able to design a proper curriculum of tasks in different outer-loop iterations from crawling, sitting, standing to walking, turning, running, just like how a human learns to walk. Instead of a curriculum, an RL objective function for intrinsic motivation also can be introduced to acquire skills ([5]), which in your perspective is “totally unsupervised” learning; we consider this work complementary to ours (we think imitations are also equally important objectives as intrinsic motivations for acquiring skills) and we’ve included the statements in the related work and the future work sections.

---

> ### Comment · AnonReviewer3 · 2019-11-15
> **Response to Rebuttal**
>
> Thank you to the authors for revising the draft and responding to the comments. The new draft is certainly more polished. However, I think there are still major issues with the method presentation and comparisons, so I am keeping my score as is.
>
> In the arguments below, the authors write "the low-level policy learns locomotions via the imitation learning with the manually set reasoning module and the internal model is trained via self-supervised learning." If this is the case, then the proposed approach amounts to using ground truth information (demonstrations and primitive labels [-1,0,1] to learn a set of motion primitives in the form of the conditioned policy, then doing planning in this high level 'primitive' space. Thus even though the internal model is learned on just its own experience, this is dependent on having a good set of primitives, which would not be possible without the imitation. So as currently formulated, it seems like the success of the method as a whole only would work given the strong supervision of demonstrations and primitive labels, so I still think calling it "self-supervised" is a stretch.
>
> However regardless of terminology - I still feel the lack of comparisons hasn't been addressed. The results still only consist of a set of ablations. There should be comparisons to approaches which use similar amounts of supervision. So given that the method is using imitation learning, there should be comparisons to methods which also use demonstrations. For example there is work on learning 'skills' from demonstrations [1] which seems especially relevant. Since the proposed approach also has labels for the skills, it should be able to outperform this style of approach. I even think showing experiments against the fully unsupervised approaches would be good, since the proposed approach should work quite a bit better, and would also better motivate using demonstrations. But without any comparisons to prior work its hard to assess how well the method works.
>
> Also minor point, the authors write "...showing powerful generalization ability of reasoning/planning process". I don't this can be claimed without quantitative numbers indicating the generalization ability is better than existing work. As it stands - the results are only for 3 tasks - left, forward, right, all of which are seen during training.
>
> [1] Roy Fox, Sanjay Krishnan, Ion Stoica, and Ken Goldberg. Multi-level discovery of deep options.
> arXiv preprint arXiv:1703.08294, 2017.

---

> > ### Author Response · Authors · 2019-11-15
> > **Response to reviewer3**
> >
> > Thanks a lot for your constructive comments.  We respect your opinion, it really helped us to figure out our contribution more clearly.
> >
> > Primitive labels $\mathbf{h}\in\{-1, 0, 1\}$ are only used for the initial policy learning ($L=1$) and such labels act just as prior for the internal model learning. For later iterations ($L>1$), a planning module then outputs any commands in $\mathcal{R}$, thus the policy is trained to perform any command in addition to $\mathbf{h}\in\{-1, 0, 1\}$. The internal model also learns a low-dimensional dynamics of such command executions. By “self-supervised” or “unsupervised”, we only meant the algorithm constructs its latent dynamics and mappings without supervision. There is no notion of “skills” in our argument and we only assume that states that a planning module should consider lie on a low-dimensional manifold and our internal model provides dynamics on that manifold in an unsupervised manner, enabling an efficient planning algorithm on that. There are many design choices for constructing internal models, and our ablation study provides a comparison between such choices. As also shown in Table 1 and 2, only the proposed framework considers both a latent state $\mathbf{z}$ and a high-level command $\mathbf{h}$, which outperforms the baseline models (many existing works are already categorized as one of those); high-dimensional states don't allow for learning exact dynamics and high-dimensional actions prohibit the effective planning.
> >
> > To show our contribution more clearly without confusion and to address your comments, we will revise our paper in the three directions.
> >
> > (1) About the terminology: We can change the title of our paper to something like “Learning to Reason: Distilling Hierarchy for Planning and Control with Internal Model” and modify the statements about self-supervised learning to be clearer (by explicitly restricting our discussion to internal model learning). Would you be satisfied with such modifications?
> >
> > (2) Discovering skills without imitation: Questions on which tasks to choose for model learning (either imitation or intrinsic motivation) are actually orthogonal to our work. Better tasks help the algorithm to find the low-dim space more quickly, but even without them, the algorithm should be able to construct an internal model as long as our assumption on a low-dim manifold is true.  To see that, we can consider the following minimal scenario: The system has a 2-dim state $s=(s_1, s_2)$ and a 2-dim action, and the dynamics is given by s’ = s+a. Suppose that a set of tasks that the agent should solve is given by those reward functions that strongly penalize differences between $s_1$ and $s_2$ and distance from the target that is only defined with $s_1$, i.e. $r=100*(s_1-s_2)^2 + (s_1-target)^2$ (the manifold $s_1-s_2=0$ here corresponds to the manifold of locomotions in the humanoid example.). Then, a lower-level policy should stabilize the states w.r.t. the manifold $s_1-s_2=0$ as well as execute commands, while a high-level planner only needs to plan state trajectories on that manifold with the reduced DOF along the manifold. That means, our framework will learn an internal model with a 1-dim latent state a 1-dim command; the internal model would think deviations of the states from the manifold (which have stable dynamics) as nothing but noise. We will include this toy example in the final manuscript.
> >
> > (3) Quantitative results: Section 4.2 shows the hierarchical policy learned with imitation directly can be applied to navigation tasks, and by generalization, we meant it is because of the generalization ability of having an internal model. In the final manuscript, we will also include the quantitative results/analysis of this example to support our claim.

---

### Official Review · AnonReviewer1 · 2019-10-23
**Official Blind Review #1**

**Rating:** 3

**Review:**

This paper proposes a latent variable model to perform imitation learning. The authors propose the model in the control-as-inference framework and introduce two additional latent variables: one that represents a latent state (z) and another that represents a latent action (h). For the generative model, the authors use a sequence latent variable model. For inferring the latent action, the authors use a particle filter. For inferring the states, the authors use an "Adaptive path-integral autoencoder," though it was unclear where the controls "u" come from. (I assume u is the same as the actions, at which point inferring the states amounts to rollout the policy in the sequence latent variable model). The authors compare to not having the latent states and/or not having the latent actions, and demonstrate that they get better imitation learning scores.

Overall, I found the paper difficult to follow and some of the reasoning a bit unclear. The experiments seemed limited in scope, given that the authors discuss reinforcement learning in general, but only provide results on the reconstruction error when doing imitation learning. It is also unclear to me whether the gains from the experiments are from their model, or from the fact that their model probably has more parameters since it has more components. It would be good for the authors to compare to existing work that uses sequential latent variables models for deep RL, such as [1,2,3]

More detailed comments:

It would be good for the authors to substantiate statements like, "Since training sas? is just a simple supervised learning problem, it had the lowest reconstruction error but the computed action from such the internal model couldn’t make the humanoid walk." with plots.

The statement, "zaz' also failed to let the robot walk, because reasoning of the high-dimensional action can’t be accurate enough." seems similarly unjustified. If the authors wanted to test this, they could train a zaz' model with some low-dimensional action (e.g. left, straight right) and verify that this works.

While the authors state that "h can be interpreted as high-level commands," but if it is inferred at every time step, why is this a "high-level" command?

Nit-picks:
- "The procedure consists of outer internal" --> "The procedure consists of *an* outer internal"
- "via via"
- "Such the sophisticated separation was"

[1] Danijar Hafner et al. Learning Latent Dynamics for Planning from Pixels.
[2] Maximilian Igl et al. Deep Variational Reinforcement Learning for POMDPs.
[3] Alex Lee at al. Stochastic Latent Actor-Critic: Deep Reinforcement Learning with a Latent Variable Model.

**Experience Assessment:**

I have published one or two papers in this area.

**Review Assessment: Checking Correctness Of Derivations And Theory:**

I assessed the sensibility of the derivations and theory.

**Review Assessment: Checking Correctness Of Experiments:**

I carefully checked the experiments.

**Review Assessment: Thoroughness In Paper Reading:**

I read the paper at least twice and used my best judgement in assessing the paper.

---

> ### Author Response · Authors · 2019-11-11
> **Initial response to reviewer1**
>
> Thank you for the constructive comments. There seem to be some misunderstandings because of the uncleanness of our initial draft. We’ve extensively revised the manuscript and let us clarify some points here as well:
>
> - This work proposes a framework to train a hierarchical policy for multi-task RL problems (imitation learning is considered just to learn such a policy), where a high-level reasoning module plans the agent’s motion and a low-level policy generates actual control inputs. We introduce a latent state, $\mathbf{z}$, and a latent command, $\mathbf{h}$, with which the reasoning problem can be formulated. A variational distribution over $\mathbf{h}$, which the reasoning module tries to compute, is parameterized by $\mathbf{u}$ that acts to the same subspace as $\mathbf{h}$ in the latent space. Therefore, the reasoning becomes nothing but the inference problem of low-dimensional $\mathbf{u}$ with the low-dimensional stochastic dynamics of $\mathbf{z}$.
>
> - The experiments show that one hierarchical policy learned by the proposed framework can perform various tasks such as imitations of human locomotion and navigations through complex environments. The ablation study in Section 4.1 tells us that, without having either $\mathbf{h}$ or $\mathbf{z}$ (which is the case of most HRL frameworks), the reasoning module can’t predict its future properly or the plan can’t be executed well by the low-level policy. Section 4.2 demonstrates that the policy trained for imitation directly can be applied to the unseen (navigation) tasks without any additional training, which shows the generalization power of reasoning.
>
> - Regarding the citations, we consider [1] closely related to our work so we’ve heavily mentioned this work in the manuscript. Thank you for letting us know. The focus on the others [2,3] is somewhat addressing partial observability rather than planning, so we haven’t included these works.
>
> ==============================================================================================================================
> Reply to the detailed comments:
> Re: about ‘sas’ and ‘zaz’
> After training the internal models of $sas’$ and $zaz’$, the reasoning module should search over the 36-dimensional action space with the limited number of particles (1024 in our case). It is infeasible due to the curse of dimensionality and, given the fact that the humanoid has very unstable dynamics, the inaccurately planed action can’t make the humanoid walk. We’ve added these comments and some figures in the experiment section and the appendix. Please watch the supplementary video as well if you haven’t seen yet: https://bit.ly/2rwIfQn
>
> Re: about $\mathbf{h}$ as high-level commands
> We couldn’t mention it in the draft, but the high-level reasoning is actually operated in a coarser time scale than the low-level policy and $\mathbf{h}$ is inferred less frequently (not every time step).
>
> Re: about Nit-picks
> Thanks for pointing those out. We fixed typos and will revise further for the final manuscript.
>
> ==============================================================================================================================
> We would be happy to discuss more and please consider to re-score the submission if you agree things become clearer.
>
> [1] Danijar Hafner et al. Learning Latent Dynamics for Planning from Pixels.
> [2] Maximilian Igl et al. Deep Variational Reinforcement Learning for POMDPs.
> [3] Alex Lee at al. Stochastic Latent Actor-Critic: Deep Reinforcement Learning with a Latent Variable Model.

---

### Author Response · Authors · 2019-11-15
**Final comment to the reviewers and AC**

We are really grateful for the reviewers' valuable and constructive comments/discussions. The followings are the issues raised by the reviewers and our responses.
==============================================================================================
- The reviewers mentioned that our draft was hard to follow, mainly because the experiment section was missing several details. We've revised the whole manuscript to clearly show our contribution and included all the experiment details. We will also release the code with the final manuscript for the reproducibility issue.

- The reviewers pointed out the lack of comparison with the existing works. As our main contribution is a method to construct an internal model, we considered the baseline models with the different design choices of a model and now, in Section 4.1 and with Table 1, we clearly state the correspondence of each choice to the existing works. That being said, we'd say the ablation study in Section 4.1 shows comparisons of our framework with the existing works, providing valuable implications and showing the novelty of this work, as the reviewer2 agreed.

- The reviewer3 worried that the usage of the terminology of "self-supervised" might cause confusion (i.e., self-supervsied learning of skills vs. self-supervised learning of a latent model).  We agree with his/her opinion and we will modify the title and the statements about that properly.

- Based on the reviewer3's comment, We will include the quantitative results for Section 4.2 in the final manuscript, to show the generalization ability of an internal model more clearly. The reviewer 2 and 3 had a question whether this learning is still valid even without imitation tasks; we will conduct the additional experiments on this for the final manuscript.
============================================================================================
Thanks again for all the comments and we hope the revisions have made our work more solid.

---

### Decision · Program_Chairs · 2019-12-19

**Decision:**

Reject

**Comment:**

The authors present a self-supervised framework for learning a hierarchical policy in reinforcement learning tasks that combines a high-level planner over learned latent goals with a shared low-level goal-completing control policy.  The reviewers had significant concerns about both problem positioning (w.r.t. existing work) and writing clarity, as well as the fact that all comparative experiments were ablations, rather than comparisons to prior work.  While the reviewers agreed that the authors reasonably resolved issues of clarity, there was not agreement that concerns about positioning w.r.t. prior work and experimental comparisons were sufficiently resolved.  Thus, I recommend to reject this paper at this time.